# Hydrolytic vs. Nonhydrolytic Sol-Gel in Preparation of Mixed Oxide Silica–Alumina Catalysts for Esterification

**DOI:** 10.3390/molecules27082534

**Published:** 2022-04-14

**Authors:** Atheer Al Khudhair, Karim Bouchmella, Pierre Hubert Mutin, Vasile Hulea, Olinda Gimello, Ahmad Mehdi

**Affiliations:** 1ICGM, University Montpellier, CNRS, ENSCM, 34095 Montpellier, France; karim.bouchmella@umontpellier.fr (K.B.); hubert.mutin@umontpellier.fr (P.H.M.); vasile.hulea@umontpellier.fr (V.H.); olinda.gimelle@umontpellier.fr (O.G.); 2Department of Chemistry, College of Science, University of Kerbala, Karbala 56001, Iraq

**Keywords:** sol-gel, catalysis, silica–alumina

## Abstract

The development of green and sustainable materials for use as heterogeneous catalysts is a growing area of research in chemistry. In this paper, mesoporous SiO_2_-Al_2_O_3_ mixed oxide catalysts with different Si/Al ratios were prepared via hydrolytic (HSG) and nonhydrolytic sol-gel (NHSG) processes. The HSG route was explored in acidic and basic media, while NHSG was investigated in the presence of diisopropylether as an oxygen donor. The obtained materials were characterized using EDX, N_2_-physisorption, powder XRD, ^29^Si, ^27^Al MAS-NMR, and NH_3_-TPD. This approach offered good control of composition and the Si/Al ratio was found to influence both the texture and the acidity of the mesoporous materials. According to ^27^Al and ^29^Si MAS NMR analyses, silicon and aluminum were more regularly distributed in NHSG samples that were also more acidic. Silica–alumina catalysts prepared via NHSG were more active in esterification of acetic acid with n-BuOH.

## 1. Introduction

Esterification of carboxylic acids is one of the major reactions in organic synthesis [1,2]. In addition to this, esterification of acetic acid with n-butyl alcohol is commercially significant because the product n-butyl acetate is widely consumed in the manufacture of artificial perfumes, photographic films, plastics, safety glasses [3], and food additives; as a solvent for coating resin; and as a pharmaceutical intermediate. In recent years, consumer demand for n-butyl acetate has grown, resulting in higher prices and profit margins. Therefore, esterification reactions have been carried out on a large scale using both homogeneous and heterogeneous catalysts. For economic purposes, homogeneous catalysts are commonly used in industries. However, these catalysts are difficult to recycle and must be neutralized, which adds cost and pollution [4]. As a result, significant research efforts have been directed to the development of heterogeneous processes to take advantage of easy separation of catalyst and products [5].

Mesoporous metal oxides and mixed oxides with acidic, basic, and/or redox properties are well-known materials used in heterogeneous catalysis, including environmental catalysis. Among these mixed oxides, silica–alumina oxides have been extensively used in esterification reactions because of their high surface area and high thermal stability [6,7,8].

Even if there is no consensus on the esterification mechanisms occurring on different (although similar) solid acid catalysts, most of the literature studies align with a mechanism that involves the Brönsted acid sites of acidic heterogeneous catalysts [9].

The strong Brönsted acidity of silica–alumina is attributed to hydroxyl groups in the “mixed phase Si-O-Al”. A debate still exists on the structure of these sites in amorphous materials (Figure 1) [10,11,12]. The Si-O-Al oxo bridge is more negatively charged than the oxygen of an Al-OH or Si-OH. Species (a) and (b) are therefore likely to generate stronger acidity.

The hydrolytic sol-gel (HSG) process, based on the formation of inorganic matrices by hydrolysis and polycondensation of molecular precursors (usually silicon or metal alkoxides), has been extensively used to prepare mixed oxide catalysts with high surface area and ordered mesoporosity [13]. This process can be defined as the conversion of a precursor solute into an inorganic solid through water-induced inorganic polymerization reactions, which can be carried out under simple experimental conditions and at low temperatures. It is one of the most popular approaches used to produce oxide materials with a high degree of control over their textural and surface properties, as well as high purity and homogeneity [14]. The nonhydrolytic sol-gel (NHSG) is another powerful technique for designing oxides and mixed oxides [15]. It is based on the reaction of chloride precursors with an oxygen donor, giving good control over the stoichiometry and homogeneity of the mixed oxide gels. The NHSG process involves condensation reactions in nonaqueous media, which significantly affects the texture, homogeneity, and surface properties of the resulting materials. It has been successfully applied to the synthesis of mixed oxides, which have shown excellent catalytic performances in various reactions [8,16].

In this work, we studied SiO_2_-Al_2_O_3_ materials prepared via both HSG and NHSG routes. The structure, texture, and acidity of the materials obtained by these procedures were evaluated and compared. Additionally, the catalytic behaviors of the mixed oxides in the esterification reaction between acetic acid and n-butanol (n-BuOH) are discussed.

## 2. Results and Discussion

### 2.1. Synthesis and Characterization of Materials

EDX analysis was used to determine the experimental compositions of the samples. As shown in Table 1, experimental weight percentages were close to the nominal ones, based on the amounts of reactants, indicating that all of the Si and Al atoms were included in the final oxides.

It’s worth noting that similar reproducible results were observed for ternary Ni/silica–alumina mixed oxide catalysts prepared by NHSG [17].

N_2_-adsorption–desorption measurements were realized and various types of isotherms were obtained, showing that the pore structure was closely related to sample composition and sol-gel route (Appendix A).

The texture of the samples depended on Si/Al ratio and on the synthetic route. As shown in Table 2, the average pore diameter (Dp) ranged between 1.8 and 13.8 nm, demonstrating the mesoporous structure of the Si_x_Al_y_ samples.

For alumina-free compositions, samples prepared via NHSG showed higher textural properties (S_BET_, Vp, and Dp) than those obtained via HSG. An increase of Al loading in the composition led to a decrease in values for NHSG samples. The specific surface area, pore volume, and pore diameter of Si_75_Al_25_ and Si_50_Al_50_ prepared in basic HSG medium were higher than those prepared in acidic conditions. The same observation was made by Agliullin et al. when they prepared porous silica–alumina using sol-gel processes at different pH values [18]. Si_25_Al_75_ and silica-free compositions prepared in acidic HSG exhibited higher textural properties than the others.

SEM images were obtained to study the morphology of the prepared materials (Appendix A). After sample grinding, the grain size was heterogeneous and varied from 800 nm to 40 μm for HSG samples and from 1 to 100 µm for NHSG compositions. Independently of the sol-gel route, solids were hard and compact for high SiO_2_ loadings (up to 75 wt%). ^29^Si and ^27^Al solid-state NMR gave information on the structure of the silica–alumina network. As shown in Figure 2, the ^29^Si CP-MAS NMR spectra of Si-containing mixed oxides showed broad resonances typical of amorphous materials [19].

The chemical shift of the Si atoms in amorphous aluminosilicates depends on the nature of the second neighbors. In amorphous silica or silica–alumina, each replacement of a (OSi) group in a Si(OSi)_4_ tetrahedron by an (OAl) or an (OH) group leads to a downfield shift of about 5 to 10 ppm [20]. The spectrum of Si1_00_Al_0_ shows three broad signals at ≈−110, −101, and −93 ppm, attributed to Si(OSi)_4_ (Q^4^), Si(OSi)_3_(OH) (Q^3^), and Si(OSi)_2_(OH)_2_ (Q^2^) sites, respectively [21]. Similar ^29^Si NMR spectra with a major Q^3^ resonance have been reported for mesoporous silicas or silica–alumina with a low Al content [20,22]. A Q^4^ signal was only observed in the Si_100_Al_0_ sample prepared via NHSG demonstrating a higher degree of condensation when using this route. For all the other samples, the incorporation of a large amount of aluminum in the silica network led to a significant low-field shift of the resonances, indicating extensive formation of Si-O-Al bridges in Si(OSi)_4−x_(OAl)_x_(Q^4^_xAl_) sites or Si (OSi)_3−x_(OH)(OAl)_x_ (Q^3^_xAl_) sites [19,23]. Substitution of Si with Al in Si(OSi)_4_ resulted in a shift in resonance of 5 ppm per Al. Since the specific surface areas of silica and silica–alumina are almost comparable, the shoulder peaks suggest the presence of a large number of Si-O-Al bonds [23]. A higher low-field shift was observed for Si_25_Al_75_ and Si_50_Al_50_ samples prepared via NHSG than those synthesized via HSG, suggesting a better distribution of Al atoms in the silica matrix.

The ^27^Al solid-state MAS NMR spectra exhibited broad resonances around 60 ppm, 35 ppm, and 5 ppm, ascribed to tetra-coordinated (Al^IV^), penta-coordinated (Al^V^), and hexa-coordinated (Al^VI^) aluminum sites (Figure 3) [23,24].

It is well known that the coordination of aluminum in amorphous silica–aluminas has an impact on their acidic properties and catalytic performance. The signal attributed to Al^IV^ sites shifted up-field when the SiO_2_ loading increased from 0 to 75%; this effect has been previously ascribed to the decreased concentration of aluminum atoms around Al sites [18,25,26]. Williams et al. attribute this effect to geometric factors (such as distorted Si-O-Al angles) and the environment of the Al^V^ and Al^VI^ sites [12]. The NHSG samples exhibited higher intensity of the peak attributed to AI^V^ sites than those ones prepared via HSG. Moreover, when Al_2_O_3_ loading increased in the NHSG samples, the proportion of Al^V^ sites increased to the detriment of Al^IV^ sites. Rarely observed in silica–aluminas, the Al^V^ species are potential unsaturated Al species like Al^IV^, and thus could be considered available for Brönsted acid site formation [27].

The structure of Si_x_Al_y_ materials depended on the synthesis route (Figure 4).

Except for alumina compositions, all XRD pattern corresponded to the PDF card standard for silica–cristobalite phase ICDD # 00-001-0424. Diffraction patterns showed similarities in profile; peak diffraction broadened at around 2θ = 22–24° as occurs in the diffraction pattern of the silica amorphous phase, but reflections were observed at 2θ = 37.3°, 43°, and 67° (Figure 4A,B) due to the presence of the γ-alumina phase in the Si_0_Al_100_ samples prepared via HSG routes (JCPDS reference no. 00-010-0425 in the International Centre for Diffraction Data database, Appendix A).

Independently of the synthesis medium, all samples showed broad peaks at 2θ = 23° attributed to amorphous silica [25]. Decreasing SiO_2_ loading in the composition of the sample led to a decrease in the intensity of this peak. No broad reflections typical of bulk γ-alumina were detected for the NHSG samples.

Temperature programmed desorption of ammonia (NH_3_-TPD) was used to examine the acidity of the Si_x_Al_y_ samples. The total amount of desorbed NH_3_ molecules per gram of sample at the end of the TPD experiment is given in Figure 5.

The relative amounts of weak and medium–strong sites were estimated using the method of Katada et al. [26]. Independently of the sol-gel route, the amount of desorbed NH_3_ roughly increased with the Al loading (up to 75 wt% for HSG samples), as reported previously for Ni-Si-Al catalysts prepared via NHSG [17] and silica–alumina materials prepared via HSG [28].

The strong acid sites could be attributed to the bridging hydroxyl groups (Si–OH–Al) and the weak acid sites could be attributed to the silanol groups or extra framework aluminum species.

Starting from 25 wt% Al_2_O_3_, the Si_x_Al_y_ samples prepared via NHSG showed higher acidity in terms of total desorbed NH_3_ and medium–strong acidity than those prepared via HSG. Obviously, both the total acidity and the strength of acid sites was affected by Si/Al ratio.

### 2.2. Esterification Behaviour of Si_x_Al_y_ Catalysts

The silica–alumina mixed oxides synthesized via sol-gel routes were evaluated in an esterification reaction, which was used as a model reaction to correlate it to their physicochemical properties (Figure 1).

Under these conditions, byproducts were not detected and the selectivity for butyl acetate product was 100%. The activity was dependent on the synthesis route of the catalyst and was markedly improved when an NHSG silica–alumina was used instead of an HSG one (Figure 6).

Starting from 25 wt% Al_2_O_3_, the ester yields obtained with NHSG catalysts were up to three times higher than those obtained with HSG samples. The HSG samples’ activities appeared not to be dependent on the composition and were similar to the reference one (“blank” without catalyst), suggesting that HSG materials were not effective in these experimental conditions. The NHSG catalysts’ activities appeared to be dependent on the alumina content. The high activity of NHSG catalysts compared to HSG ones can be clearly explained by the acid effect of the Si-O-Al oxo bridges. The NHSG samples exhibited both higher acidity and better dispersion of Al atoms in the amorphous silica matrix than those prepared with HSG. It is well known that solid catalysts which are acidic exhibited good catalytic performance in esterification reaction [29]. To take into account the differences in Al_2_O_3_ loading of the catalysts, we investigated the esterification reaction using a fixed Al mole percentage (20 mol% of Al related to acetic acid moles). As expected, the HSG samples were virtually inactive (Figure 6B).

The Si_50_Al_50_ and Si_75_Al_25_ catalysts showed the highest activities, with an ester yield of 53% (Figure 6A) and 44% (Figure 6B), respectively.

## 3. Experimental: Materials and Methods

### 3.1. Preparation of the Catalysts

The SiO_2_-Al_2_O_3_ mixed oxides with different Si/Al ratios (0, 0.3, 1, 2.9, and ∞) were synthesized using as molecular precursors tetraethylorthosilicate TEOS (Si(OC_2_H_5_)_4_ 98% from Sigma Aldrich) silicon tetrachloride (SiCl_4_ 99.98% from Alfa Aesar), aluminum trichloride (AlCl_3_ 99.98% from Alfa Aesar), aluminum isopropoxide (Al(OCH(CH_3_)_2_)_3_ 98% from Sigma Aldrich), and aluminum nitrate (Al(NO_3_)_3_ 95% from Sigma Aldrich). Ethanol absolute and distilled water were used as solvents. Dry diisopropyl ether (iPr_2_O, H_2_O < 4 ppm) and CH_2_Cl_2_ (H_2_O < 6 ppm), used as oxygen donor and solvent, respectively, were purchased from Aldrich with 99% purity and were further dried using an inert solvent purification system (Pure-solv line of solvent purification systems from Innovative Technology).

HSG synthesis. A colloidal SiO_2_ sol was prepared using TEOS, ethanol, and water (1:1:2 volume ratios) in acidic medium (pH of 1.5, adjusted with HCl). The solution was stirred for 1 h at room temperature to give a transparent sol. An Al_2_O_3_ sol was prepared following Yoldas’ method [30,31]. Al(OCH(CH_3_)_2_)_3_ (0.1 mol) was dissolved in distilled water (pH 1.5) that had been preheated to about 90 °C. In addition, the solution was stirred for 1 h to obtain a homogenous sol. By mixing and stirring the SiO_2_ and Al_2_O_3_ sols, the SiO_2_-Al_2_O_3_ sol was prepared. The sol was heated to 50 °C for 3 days (aging) and the gel was washed under air atmosphere with ethanol, acetone, and diethyl ether successively and dried at 25 °C under vacuum (10 Pa) for 1 h and then for 4 h at 120 °C. The resulting xerogel was crushed in a mortar and calcined in air at 500 °C for 5 h (heating rate 10 °C min^−1^). The SiO_2_-Al_2_O_3_ samples were also prepared via HSG in basic medium, at a constant pH of 8.5. The SiO_2_ sol was prepared by mixing TEOS with distilled water (1:2), which was adjusted to a pH of 1.6 with nitric acid. The TEOS was then allowed to hydrolyze overnight at room temperature under continuous agitation to give a homogeneous sol. Saturated aluminum nitrate was added (the quantity depended on the desired amount of Al_2_O_3_). Mixing was continued for 1 h. Finally, ammonium hydroxide solution (2 M) was added dropwise to form a hydrogel at pH 8.5. The gel was washed with distilled water and ethanol. The xerogel was dried at 25 °C under vacuum (10 Pa) for 1 h at room temperature and then for 4 h at 120 °C. The resulting xerogel was crushed in a mortar and calcined in air at 500 °C for 5 h (heating rate 10 °C min^−1^).

NHSG synthesis. SiO_2_-Al_2_O_3_ mixed oxides were synthesized under argon atmosphere (inside a glove box), starting from SiCl_4_ and AlCl_3_ as silicon and aluminum precursors. Dry diisopropyl ether was used as the oxygen donor and all the reactants were dissolved in 10 mL of dry CH_2_Cl_2_ inside an autoclave under stirring. The reaction was performed at 110 °C under autogenous pressure for 4 days. Colorless or white gel products were obtained. The gels were isolated by filtration and purified by washing with CH_2_Cl_2_ to remove unreacted species and byproducts. The final products were dried at room temperature for 1 h and then dried at 100 °C for 4 h. The xerogels were crushed in a mortar and calcined in air at 500 °C for 5 h (heating rate 10 °C min^−1^). Due to the slow kinetic reaction, AlCl_3_ (0.015 mmol) was used as the catalyst to obtain a gel for the preparation of the alumina-free mixed oxide (Si/Al = ∞).

The samples were labeled Si_x_Al_y_, where x and y represent the expected SiO_2_ and Al_2_O_3_ weight loadings, respectively.

### 3.2. Characterization of the Catalysts

The atomic percentages of Si and Al were obtained via energy dispersive X-ray spectroscopy (EDX). Measurements were carried out using an X-Max Silicon Drift detector mounted on an FEI Quanta FEG 200 scanning electron microscope. To obtain reliable results in the analyses, the data for each sample are the averages of three separate measurements. The use of a piece of Scotch^®^ to fix the powder and the partial vacuum (0.3 Torr) in the measuring chamber did not allow for a correct analysis of the concentrations of C and O. X-ray powder diffractograms (XRD) were obtained on a Philips X-pert Pro II diffractometer using Kα copper Cu radiation (λ = 1.5418 Å) as a radiation source. The diffractograms were obtained in the range of 2θ from 20 to 80° in steps of 0.02°. The solid-state ^29^Si CP-MAS NMR spectra were recorded on a 59.6 MHz VARIAN VNMRS 300 spectrometer with a MAS (magic angle spinning) probe and 7.5 mm zirconia rotors, using a cross-polarization sequence (CP) with a 3 ms contact time, a 5 s recycling delay, and a 5 kHz spinning frequency. Chemical shifts were referenced to tetramethylsilane using octa(dimethylsiloxy)silsesquioxane Q_8_M^H^_8_ as a secondary reference. The solid-state ^27^Al NMR spectra were recorded on a 104.26 MHz VARIAN VNMRS 400 spectrometer with a MAS (magic angle rotation) probe with 3.2 mm zirconia rotors, using a single pulse sequence with 1H decoupling, pulses of 1 μs (corresponding to a pulse angle of π/12), a recycling time of 1 s, and a spinning frequency of 20 kHz. The chemical shifts were referenced with respect to an aqueous solution of aluminum nitrate. N_2_ physisorption experiments were performed at 77 K on a Tristar instrument from Micrometrics. The calcined samples were outgassed overnight at 150 °C under vacuum (2 Pa). The specific surface area (S_BET_) was determined using the BET method in the 0.05 to 0.30 P/P_0_ range. The total pore volume V_P_ was measured at P/P_0_ = 0.985. The average pore diameter D_P_ was calculated from the total pore volume and the specific surface area (D_P_ = 4 V_P_/S_BET_). The pore size distribution was derived from the desorption branch using the BJH method. The total volume of the micropores of the samples was estimated by the t-plot analysis. The acidity was evaluated using temperature programmed desorption of ammonia (NH_3_-TPD) on a Micromeritics AutoChem 2910 apparatus with a thermal conductivity detector. The sample (100 mg) was preheated in helium at 200 °C for 60 min (heating rate 10 °C min^−1^). Adsorption of NH_3_ (pure ammonia, flow rate 30 mL min^−1^) was done at 100 °C for 60 min. Physisorbed NH_3_ was removed by purging with helium at 100 °C for 1 h (flow rate 30 mL min^−1^). The TPD measurement was conducted by heating the sample from 100 to 650 °C with 10 °C min^−1^ as the heating rate. The relative amounts of weak and medium–strong sites were estimated using the method described by Katada et al. [26]. A Gaussian function centered at approximately 180 °C was fitted to the peak and attributed to weak acid sites. The rest of the peak was attributed to medium and strong acid sites. The products of the esterification reaction were analyzed using a gas chromatograph with flame-ionization detection GC-FID (Agilent 6850). The chromatographic separation was performed on an HP-1 capillary column (30 m × 0.25 mm × 8 µm). Hydrogen carrier gas was set at a linear velocity of 24 cm s^−1^. The temperature of injector and detector was set at 250 °C with a split ratio of 30 for the injector. The oven temperature program was set at an initial temperature of 40 °C for 4 min and was then increased to 100 °C at 8 °C min^−1^ for 1 min; the final temperature was 110 °C (20 °C min^−1^).

### 3.3. Catalytic Tests

The catalytic activity in the esterification reaction (acetic acid and n-butanol) was studied in a 50 mL plastic centrifuge tube at 80 °C for 1 h. Prior to reaction, the mixed oxides were activated by heating under vacuum at 150 °C for 5 h. A stock solution was prepared by adding acetic acid (6.00 g, 0.1 mol, VWR chemicals 99–100%) and 1-butanol (14.80 g, 0.2 mol, Acros Organics 99.5%). Toluene (Fisher chemical 99.8%) was added to obtain a 100 mL total volume. The esterification reaction was carried out by adding 5 mL of the stock solution to the catalyst. For each catalyst sample, two tests were realized: a fixed mass of catalyst (100 mg) and a fixed mole percentage of Al. The mass weighting of the catalyst was calculated to obtain 20 mol% of Al related to the number of mole of acetic acid. A reference reaction was also carried out in the absence of sample (labeled “blank”). After 1 h at 80 °C, the tubes were cooled down to room temperature. The tubes were centrifuged at 20,000 rpm for 20 min. The supernatants were analyzed using GC. The ester yield (butyl acetate) was defined as the number of moles of ester produced per moles of acetic acid introduced. Byproducts were not detected.

## 4. Conclusions

In this work, we showed that HSG and NHSG routes were successfully used to prepare Si_x_Al_y_ mixed oxides. For HSG, the binary mixed oxides were prepared via a multistep process in acidic and basic media. For NHSG, the one-step ether route was used to prepare samples. We showed that independently of the sol route, Si/Al ratio can be easily controlled, knowing that it played an important role in the structure, texture, and acidity of the samples. The mesoporous and amorphous alumina-rich mixed oxides prepared with NHSG showed higher acidity and Al^V^ species than those prepared with HSG. Silica–aluminas and alumina samples prepared with NHSG were highly active in the esterification reaction performed at 80 °C. This comparison study clearly points out the potential of NHSG for the preparation of mixed oxides and open the doors for the design of tunable catalytic materials.

## Data Availability

Not applicable.

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
