# Peer review of "Hydrolytic vs. Nonhydrolytic Sol-Gel in Preparation of Mixed Oxide Silica–Alumina Catalysts for Esterification"

_molecules, 2022, doi:10.3390/molecules27082534_

Round 1
Reviewer 1 Report
he authors reported the synthesis of Si50Al50 and Si75Al25 catalysts by NHSG method, which shows considerable activity for the esterification of acetic acid with n-BuOH. Here are some suggestions to be considered before the publication of this work:
(a) Lines 95-99: the specific surface area (SBET), pore volume (Vp) and pore diameter do not completely conform to the conclusion described in the article. Si50Al50 and Si0Al100 prepared by acidic HSG showed lower pore volume (Vp) than those obtained following NHSG. Si50Al50 and Si75Al25 prepared by acidic HSG showed lower pore diameter than those obtained following NHSG. Si0Al100 prepared by HSG showed a lower pore diameter than those obtained following NHSG. The specific surface area and pore diameter of Si25Al75 and Si0Al100 prepared in a basic HSG medium are lower than those prepared in acidic conditions.
(b) In the Results and Discussion section, if the XRD patterns in your article can be added to the corresponding pdf cards of γ-Al2O3 and SiO2, I think it can be more clearly seen what you want to express. Secondly, although the preparation method corresponding to each of your pictures can be understood from the context, I think it may be better if it can be further explained in the annotations of the pictures. Another small question is whether the C in your picture is capitalized, please confirm it when it is convenient for you.
(c) In Figure 5, The influence of Si/Al ratio on the total acidity and acid strength of samples prepared by basic HSG is not regular, which is not consistent with your description in line 166.
(d) In Figure 6, you compare the activity of the catalyst prepared by the NHSG method with the catalyst prepared by the HSG method, and found a very significant increase in activity, but under either acidic or basic conditions, The activity of the catalyst prepared by the HSG method is very similar to that of the blank group, so I think if you can find the activity of other heterogeneous catalysts that are widely used in the industry and compare them, you may get more recognition.
(e) In lines 187-188, you attributed the high activity of catalysts prepared by NHSG to the stronger acidity and higher Al dispersion. But the acidity of catalysts prepared by basic HSG and acidic HSG are different, they have the same performance in all cases.
(f) In lines 311-313 of the conclusion, you mentioned that the catalyst prepared by the NHSG method showed better activity at 80°C. I think if you could add an experiment using the best catalysts screened in the activity section above, doing the activity at other temperatures might give the data at this temperature more credible.
Author Response
See attached cover letter

Reviewer 2 Report
This paper reports the study of performed the development of green and sustainable materials as heterogeneous catalyst is a growing area of research in chemistry. In this paper, mesoporous SiO2-Al2O3 mixed oxides catalysts with different Si/Al ratio were prepared by hydrolytic (HSG) and non-hydrolytic sol-gel (NHSG) processes. The HSG route was explored in acid and basic media, while NHSG was investigated in the presence of diisopropylether as oxygen donor. The obtained materials were characterized by EDX, N2-physisorption, powder XRD, Si, Al MAS-NMR and NH3-TPD. This approach offers good con- trol of composition and the Si/Al ratio was found to influence both the texture and the acidity of the mesoporous materials. According to Al and Si MAS NMR analyses, silicon and aluminum were more regularly distributed in NHSG samples that were also more acidic. Silica-alumina catalysts prepared by NHSG were more active in esterification of acetic acid with n-BuOH The experimental results are interesting and informative, but some data were not well presented. Details are listed below.
- Authors synthesized mesoporous SiO2-Al2O3 mixed oxides catalysts.The author presented the compositional analysis of metals with EDS. However, since EDS is not accurate, it is necessary to add data by analyzing it with ICP-AES..
- The author need to add data confirmed though EDS mapping to determine whether it is alloy of SiO2-Al2O
- The authors synthesized the catalyst, but could not find data on its size and mesoporous size. Therefore, the author should add TEM or SEM data and add the average size and mesoporous size.
Author Response
See attached cover letter

Round 2
Reviewer 1 Report
The authors addressed my questions well.
Reviewer 2 Report
Good